

# Exclusions for resolving urban badger damage problems: outcomes and consequences

Alastair I. Ward[1,2], Jason K. Finney[2], Sarah E. Beatham[2], Richard J. Delahay[3], Peter A. Robertson[2,4] and David P. Cowan[2]

[1] School of Environmental Sciences, University of Hull, Hull, United Kingdom
[2] National Wildlife Management Centre, Animal and Plant Health Agency, York, North Yorkshire, United Kingdom
[3] National Wildlife Management Centre, Animal and Plant Health Agency, Woodchester Park, Gloucestershire, United Kingdom
[4] School of Biology, Newcastle University, Newcastle-upon-Tyne, United Kingdom

## ABSTRACT

Increasing urbanisation and growth of many wild animal populations can result in a greater frequency of human-wildlife conflicts. However, traditional lethal methods of wildlife control are becoming less favoured than non-lethal approaches, particularly when problems involve charismatic species in urban areas. Eurasian badgers (*Meles meles*) excavate subterranean burrow systems (setts), which can become large and complex. Larger setts within which breeding takes place and that are in constant use are known as main setts. Smaller, less frequently occupied setts may also exist within the social group's range. When setts are excavated in urban environments they can undermine built structures and can limit or prevent safe use of the area by people. The most common approach to resolving these problems in the UK is to exclude badgers from the problem sett, but exclusions suffer a variable success rate. We studied 32 lawful cases of badger exclusions using one-way gates throughout England to evaluate conditions under which attempts to exclude badgers from their setts in urban environments were successful. We aimed to identify ways of modifying practices to improve the chances of success. Twenty of the 32 exclusion attempts were successful, but success was significantly less likely if a main sett was to be excluded in comparison with another type of sett and if vegetation was not completely removed from the sett surface prior to exclusion attempts. We recommend that during exclusions all vegetation is removed from the site, regardless of what type of sett is involved, and that successful exclusion of badgers from a main sett might require substantially more effort than other types of sett.

## INTRODUCTION

Landscapes are becoming increasingly urbanised throughout the world (*United Nations, 2003*), threatening traditional wildlife habitat. In contrast, protective wildlife legislation has, most likely contributed to the growth of some wildlife populations (*Heydon, Wilson & Tew, 2010*). Together, these two factors are likely to drive an increase in the frequency

Corresponding author
Alastair I. Ward, a.i.ward@hull.ac.uk

of interactions and conflicts between humans and wildlife (*DeStefano & DeGraaf, 2003*; *Soulsbury & White, 2015*; *White & Ward, 2010*). Such conflicts can be extremely costly, for example, wildlife damage and its control in urban areas of the USA has cost an estimated $8.6 billion USD per year (*Conover, 2001*). Nevertheless, growing public antipathy towards lethal control of wild animals (*Littin & Mellor, 2005*; *White et al., 2003*) and ethical obligations regarding animal welfare require that we include the development of further humane and non-lethal techniques for intervention. Moreover, interventions need to be sustainable, such that they effectively and permanently resolve problems rather than simply delaying or moving them elsewhere (*Davison et al., 2011*), whilst also allowing the persistence of viable wildlife populations in rapidly urbanising landscapes.

Many meso-carnivores have become well-adapted to an existence in urban, suburban and peri-urban environments (hereafter referred to collectively as urban environments), including foxes (*Vulpes vulpes*; *Harris & Smith, 1987*) and stone martens (*Martes foina*; *Herr et al., 2010*) in Europe, and racoons (*Procyon lotor*; *Prange, Gehrt & Wiggers, 2003*) in North America. In particular, Eurasian badgers (*Meles meles*), are widespread and increasingly abundant in rural and urban environments across much of Eurasia (*Roper, 2010*). They have become well habituated to urban areas (e.g., England: *Davison et al., 2008*, Norway: *Bjerke, Østdahl & Klimer, 2003*; Japan: *Tanaka, Yamanaka & Katsuhiko, 2002*) and their presence is often welcomed, and sometimes actively encouraged, by householders (*Bjerke, Østdahl & Klimer, 2003*). However, their presence can result in problems such as damage to gardens and buildings, mainly through sett (subterranean burrow) excavation. In England the Protection of Badgers Act 1992 makes it an offence to interfere with any badger or their sett without a licence from the statutory authority (Natural England). Of the 500–600 applications per year for such licences received by Natural England, an increasing proportion has arisen in urban areas in some regions in recent years. For example, during 1994–1996 approximately 15–20% of licence applications came from urban areas of southern and eastern England, rising to approximately 25–40% during 2002–2004 (*Delahay et al., 2009*). Licence applications can be submitted by the landowner or someone appointed by them and any subsequent management action permitted by the licence can be undertaken by the licensee.

The potential options for managing problems caused by badger sett construction currently include doing nothing, exclusion of badgers from the problem sett, translocation or localised culling. Doing nothing may not be an acceptable option from the licence applicants' perspective and is generally only a realistic option where there is no clear evidence of serious damage. Translocation is unlikely to constitute a reliable or desirable solution in many cases since it is expensive (*Beringer et al., 2002*), has a questionable success rate (*Griffith et al., 1989*) and risks causing economic and ecological problems by, for example, impacting on local fauna and flora and spreading disease to new areas (*Craven, Barnes & Kania, 1998*; *Massei Smith et al., 2010*). Proposals to cull badgers can be very unpopular with the public. The most common approach currently used to resolve problems caused by urban badgers is to exclude them from their setts by fitting gates that open outwards but not inwards over every entrance hole, and then to destroy the sett structure once the badgers have been successfully excluded (*Delahay et al., 2009*). Advice on exclusion methods is currently based on expert opinion (e.g., https://www.gov.uk/guidance/badgers-

surveys-and-mitigation-for-development-projects, accessed 11 January 2016). Exclusion may include measures to facilitate fitting of one-way gates, such as vegetation removal, and to prevent badgers from excavating new holes at the site, for example by laying a covering of heavy-gauge galvanized chain link fencing mesh over the ground surrounding the sett. Under some circumstances, such as when badgers are to be excluded from only part of the sett, or whether doubt remains as to whether all badgers have been excluded, exclusion has been followed by careful and systematic excavation of the sett to ensure that all badgers have been successfully evicted. Such works are the responsibility of the licensee or their agent and typically cost GB£5,000–£10,000 for the exclusion of badgers from a modest-sized sett on residential property (*Davison et al., 2011*).

Many hundreds of sett exclusions are undertaken each year in England alone (*Delahay et al., 2009*), yet evidence of the effectiveness of different exclusion methods is largely lacking. Following the review of reports from those licensed to undertake exclusions, *Delahay et al. (2009)* noted a low success rate for those involving main setts in urban environments. However, sett management factors that influence the success or failure of attempts to exclude badgers have yet to be empirically demonstrated.

We aimed to evaluate conditions under which licensed attempts to exclude badgers from their setts in urban environments were successful in order to identify ways of modifying practices to improve the chances of success. Improved practices may help address the apparently low success rate of exclusions in urban England, but they may also offer options for the management of problems associated with other burrowing mammals where lethal control is a least-preferred solution. Our objective was to evaluate the factors likely to determine the outcome of licensed badger exclusions from setts.

## METHODS

The study proceeded with the approval of the Central Science Laboratory's (now part of the Animal and Plant Health Agency) Ethical Review panel and was conducted under licence 20072352 from Natural England. The broader project within which this study was conducted required the capture and marking of wild badgers under general anaesthesia (*De Leeuw et al., 2004*), which was undertaken under licences PPL 60/3351 and PIL 60/9302 from the Home Office.

For the purpose of the present study we collated information from Natural England Wildlife Advisors on 39 of 45 applications for licensed sett exclusions in urban, suburban and peri-urban (albeit within the built environment) areas throughout England during the period May 2004 to October 2010. Of these, 32 were included because it was clear that exclusion attempts were going to proceed that year; a small proportion of applications do not result in action being taken against the sett. In each case at least two visits were undertaken, the first to survey badger activity at the sett prior to the commencement of licensed activity and the second to assess exclusion methods once they had been put in place.

## Sett surveys

A full sett survey was conducted at each site, following a procedure adapted from *Wilson et al. (2003)*. At each sett the number of entrance holes was counted and each hole was scored for the condition of the tunnel floor (0 = obstructed, 1 = loose, 2 = compacted), the condition of the tunnel walls (0 = unpolished, 2 = polished), the presence of footprints in the tunnel or at the entrance (0 = absent, 3 = present), the presence of recently excavated soil on the spoil heap (0 = absent, 3 = present), and the presence of fresh bedding material on the spoil heap (0 = absent, 3 = present). The scores for each category were summed to produce an activity score for the sett, which broadly correlates with badger abundance at the sett (*Wilson et al., 2003*). The sett was also described as a main sett or other type of sett based on its physical characteristics. Main setts are those that are inhabited all year round, are used for breeding and are the focal point of the badger social group. Other types of setts include outlier setts, which are used less frequently, so are not permanently occupied, and may not be used by every member of the social group (*Neil & Cheeseman, 1996*). During the present study, main setts were assigned on the basis of their relatively large size, the presence of several active holes with sizeable spoil heaps or other signs of significant activity and if obvious badger runs were observed radiating from the sett.

## Exclusion surveys

During typical sett exclusions one-way gates are installed at entrance holes and activity is monitored for at least 21 days by placing small sticks just inside the gate, which are disturbed if a badger exits past them. When disturbance of the sticks or other evidence of activity within the sett ceases, it is considered to be unoccupied. When no signs are detected to indicate that badgers have gained entry to the sett over a period of 21 consecutive days, this is taken as evidence that badgers are no longer occupying the sett (https://www.gov.uk/guidance/badgers-surveys-and-mitigation-for-development-projects accessed 11 January 2016) and the structure may be permanently sealed or destroyed.

When the licensee confirmed that badger activity at the sett had been absent for 21 to 31 days (i.e., the 21 day exclusion period, plus up to 10 days of leeway to account for breaches back into the sett and subsequent remedy), we scored this as a successful exclusion. If the licensee confirmed that badger activity had not been prevented at the sett within the 31 days we scored this as a failure. These assessments risked over-estimating the failure rate, since improvements to exclusion practices as the exclusion progressed beyond 31 days, which we did not assess, could have resulted in a successful exclusion. Nevertheless, failure to exclude badgers within 31 days implies that the measures used initially were inadequate to achieve exclusion within that period.

We re-examined setts approximately 31 days after implementation of exclusion measures. The licensee reported whether the exclusion was intended to be complete (i.e., targeting the whole sett) or partial (i.e., targeting part of the sett). A partial exclusion might be attempted where the licensee does not have access to every entrance hole, such as if some holes are present on the property of an uncooperative neighbour. We recorded the number of active and inactive entrance holes, the number of holes fitted with one-way gates or otherwise blocked, removal of surface vegetation, the proportion of the ground

surface that needed to be covered with chain link that was covered with chain link (based on our opinion of the area of the property susceptible to badger digging and that could allow re-entry to the sett), and information on any other exclusion methods that may have been used, such as hard core-filled trenching around the site or excavation of the structure following gating. The presence and number of alternative setts used within the social group's range was identified for each case during a programme of tracking of badgers fitted with radio telemetry collars (not reported here). Collared badgers were tracked each day for two weeks prior to exclusion and two weeks during exclusions.

## Data analysis

In order to predict the likelihood of success, we modelled exclusion outcomes (success or failure) using a generalized linear model procedure, with a binomial distribution and logit link function. Variables entered into the full model to attempt to explain variation in success were: Exclusion type (1 = partial, 2 = complete), Sett type (1= main, 2 = other), Sett activity (score), Holes (1 = all holes gated, 2 = fewer than all holes gated), Vegetation removal (1 = nil, 2 = partial removal of vegetation, 3 = complete removal of vegetation), Other setts (1 = no other setts identified in range, 2 = one or more alternative setts identified in range), and Chain link (1 = incomplete coverage with chain link, 2 = complete coverage). Excavation of the sett following gating was not included as a factor in the full model since it was undertaken infrequently and was invariable i.e., all excavations were successful at excluding badgers. For this analysis we excluded two cases, which were re-attempts at exclusion at sites where similar efforts had failed during the previous season. However, we included two cases that were re-attempts at exclusion following an interlude of more than one year because the contractors employed to undertake the exclusion and the methods used varied between successive attempts. Prior to derivation of the full model, we sought to control collinearity by removing one variable from each pair of explanatory variables that were strongly correlated with one another ($r_s < 0.6$, $P < 0.05$). The remaining variables were all then entered into the model simultaneously. The least significant variable was removed and the analysis was re-run. Once only significant variables remained within the model we attempted to sequentially fit all two-way interactions to form a suite of different candidate models. The model with only significant variables and their interactions, and with the lowest Akaike's Information Criterion score corrected for small sample size (AICc) was selected as the final model (*Burnham & Anderson, 2002*).

Model fit was assessed by standard validation procedures (residuals checking, Cook's distance), and by fitting the data from the cases into the model in order to calculate the number of times that the model correctly predicted the exclusion outcome. All statistical analyses were undertaken using SPSS 17.0.0 (IBM, New York).

## RESULTS

### Sett surveys

Of the 45 sett exclusion licence applications referred to us, 39 were investigated for inclusion in the study. The 32 cases studied represented 30 sites; four cases were from two sites, each studied during two separate years. Among the 30 sites 27 were suburban, mostly in private

**Table 1  Characteristics of successful and unsuccessful outcomes in 32 cases of attempted badger exclusion.** Figures given are frequency by category, or the mean (and standard deviation).

| Characteristic | | Successful | Unsuccessful |
|---|---|---|---|
| Exclusion type: | Complete | 15 | 10 |
| | Partial | 5 | 2 |
| Sett type: | Main | 10 | 10 |
| | Other | 10 | 2 |
| Vegetation removal: | Nil | 8 | 6 |
| | Partial | 7 | 5 |
| | Complete | 5 | 1 |
| Excavation: | No | 16 | 12 |
| | Yes | 4 | 0 |
| Sett activity score (mean and SD) | | 30.8 (22.7) | 52.1 (42.2) |
| Proportions of holes gated (mean and SD) | | 0.88 (0.31) | 0.89 (0.29) |
| Proportion of chain link (mean and SD) | | 0.49 (0.44) | 0.53 (0.40) |
| Number of alternative setts (mean and SD) | | 1.95 (1.23) | 1.33 (0.89) |

gardens adjacent to housing, and three were peri-urban, one of which was a churchyard, the other two being private gardens adjacent to housing but bordering rural landscapes. Main setts were the focus of 20 of the 32 cases (Table 1). Setts were spread over more than one property in 10 cases, one of which involved four properties.

Some clustering of cases was observed in space and time. Excluding the repeat cases, five were in the south east (East Sussex; one in 2005 and two each in 2006 and 2007), seven in the midlands (East Midlands: one in 2004, West Midlands; one in 2006, two in 2007 and three in 2008), five in the east (Essex; one in 2004, three in 2005 and one in 2007), four in the north west (Cheshire; one each in 2006, 2007, 2008 and 2010), two in the south (Hampshire; 2008 and 2009), two in Greater London (2006 and 2007), two in the west (Gloucestershire; one each in 2008 and 2009) two in the North East (South Yorkshire and Northumberland, both in 2009) and one in Oxfordshire (2008) (Fig. 1). However, in only one instance we identified a direct association between two consecutive cases. In this instance, badgers marked at a sett in Essex, from which they were excluded, were trapped the following year at another sett on a nearby property.

## Exclusion surveys

Of the 32 attempts to exclude badgers from setts, 20 were successful and 12 were unsuccessful (Table 1) based on the subsequent presence or absence, respectively, of signs of badger activity for at least 21 consecutive days.

No explanatory variables were collinear, but some were weakly/moderately correlated. For example, Chain link was negatively correlated with Excavation ($r_s = -0.455$, $n = 30$, $P = 0.012$), which is consistent with licensees resorting to excavation after deploying inadequate chain link and subsequent exclusion failure. However, since no unsuccessful cases involved excavation, it was not possible to include this variable in further analyses.

The full model was not significantly different from the intercept only model ($\chi^2 = 12.56$, $df = 8$, $P = 0.128$, and AICc = 55.96). After exclusion of non-significant terms (factors,

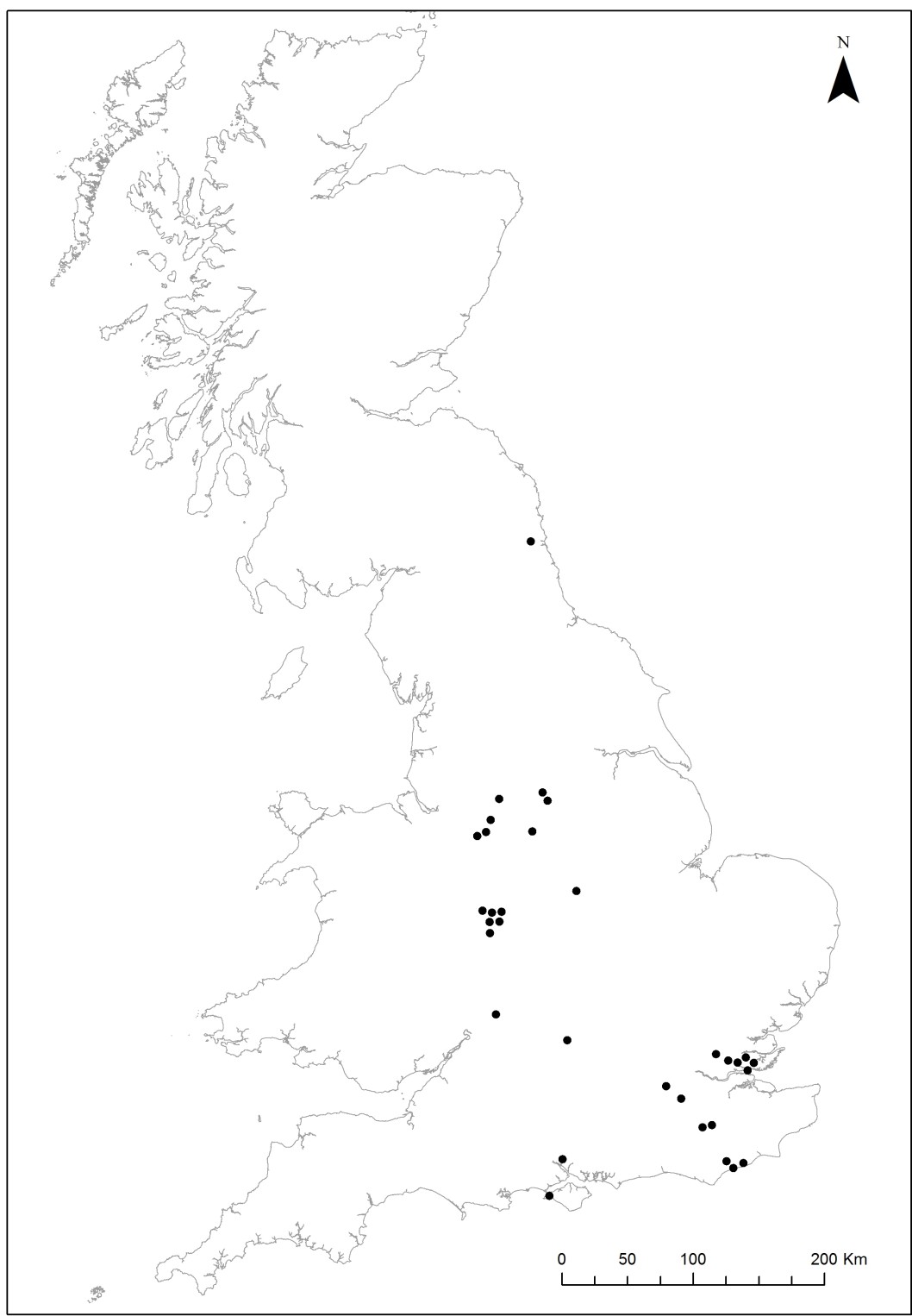

**Figure 1** **The distribution of 32 licensed sett exclusions studied.** Plotted locations may be 1 km from the study site locations to allow discrimination of clustered cases.

**Table 2** Final model to explain the failure of efforts to exclude badgers from their setts.

| Variable | $\beta$ | SE | Odds ratio | Wald | df | P |
|---|---|---|---|---|---|---|
| (Intercept) | −4.32 | 1.50 | 0.013 | 8.31 | 1 | 0.004 |
| Sett type (main) | 2.71 | 1.02 | 15.00 | 7.02 | 1 | 0.008 |
| Vegetation removal (incomplete) | 2.71 | 1.28 | 15.00 | 4.46 | 1 | 0.035 |

variables and their two-way interactions), only two factors remained. Among these Vegetation removal was significant for level 0 (no vegetation removal versus level 2 complete removal), but not for level 1 (partial removal versus complete removal, $P = 0.083$, AICc = 19.65). Consequently, we combined categories within Vegetation removal (1 = incomplete removal, 2 = complete removal) and re-ran the analysis. The final model was significantly different from the intercept only model ($\chi^2 = 10.66$, $df = 2$, $P = 0.005$ and AICc = 14.16), with two factors remaining (Table 2). Running the same analysis as a binary logistic regression revealed the moderate predictive power of the model (Cox and Snell $r^2 = 0.299$, Nagelkerke $r^2 = 0.404$), correctly predicting the outcome of cases on 80% of occasions (correct prediction of success = 83%, correct prediction of failure = 75%).

Sett exclusion using one-way gates was 15 times more likely to fail if a main sett was involved (Odds ratio in Table 2) than if it were some other type of sett and 15 times more likely to fail if all vegetation was not removed than if only some or no vegetation was removed.

## DISCUSSION

A high proportion of applications to Natural England to exclude badgers from their setts in urban areas involve main setts (*Delahay et al., 2009*). Main setts are limiting resources for badgers; in rural environments they tend to be located on slopes of free-draining soils (*Kruuk, 1978*). Moreover, they are intensely defended as a key breeding resource (*Doncaster & Woodroffe, 1993*). This may explain why the presence of alternative setts within a social group's range, which are typically fewer in urban areas in comparison with rural areas (*Davison et al., 2008*), was not significantly associated with the success or failure of attempts to exclude badgers in this study: badgers sought to remain in the main sett irrespective of alternative sett availability. However, our sample size was small and factors and variables tested were not equitably balanced between successful and unsuccessful exclusion attempts. It is possible that a larger and better balanced study than this might provide greater clarity on the importance of alternative sett availability for the successful exclusion of badgers.

In the present study main setts constituted the majority of problem cases investigated and it is perhaps no surprise that attempts to exclude badgers from them were far less frequently successful than exclusions from other types of sett. During our study badgers were successfully excluded from 50% of main setts, which is consistent with the 35% reported by *Delahay et al. (2009)*. Sett exclusion using one-way gates was more likely to fail at a main sett than at some other type of sett and more likely to fail if all vegetation was not removed. Moreover, the failure of exclusion attempts at sites at which vegetation was completely removed was almost completely explained by the presence of a main sett, and

failure at sites with other types of setts was almost completely explained by the failure to remove all vegetation. Exclusion of badgers from a main sett had an even greater likelihood of failure if all vegetation was not removed from the site. However, the significance of Vegetation removal became more evident once this factor was grouped from three into two categories. Further support for the importance of vegetation removal for the successful exclusion of badgers from their setts would require a larger sample size that was better balanced between success and failure and levels of vegetation removal.

Vegetation removal might act in two ways: it creates disturbance and removes cover from the sett, which may make the site unattractive to badgers (*Jepsen et al., 2005*; *Remonti, Balestrieri & Prigioni, 2006*; *Smal, 1995*; *Thornton, 1988*; *Wright, Fielding & Wheater, 2000*), and may also facilitate the deployment of chain link over the sett surface. A further field experiment would be required to separate the mechanisms driving the effect of vegetation removal on badger behaviour. Complete removal of vegetation at the scale of the typical English suburban garden (circa 150 m$^2$) is unlikely to cause significant long term environmental impacts.

Chain link was not significantly associated with the success/failure of sett exclusion attempts, probably because in the examples we studied it was rarely deployed securely. Indeed, in many cases where it was deployed, we believe (but have no empirical data in support) that failure to cover a sufficient area, failure to adequately secure joins between sections of chain link or to bury the periphery into a trench, resulted in badgers re-gaining access to the sett at these weak points. Our observations also lead us to believe that scrupulous deployment of chain link was more important for successful exclusion of main setts than for other sett types. However, to confirm or refute these hypotheses, further observation of cases or field experimentation are required where chain link is deployed securely and insecurely.

In three cases where badgers were not successfully excluded from setts, post-gating excavation was undertaken following failure of the one-way gates over a period of at least 21 days. In all three cases, the excavation was successful in removing the badgers. This practice resulted in the destruction of the sett, and hence was particularly likely to succeed, although our sample size was too small to allow general conclusions. Moreover, while excavation may be effective, the cases observed during this study required substantial effort, in terms of human resources and the skilful capture of badgers remaining in the sett at the time of excavation.

A further challenge associated with main setts in urban areas is that they may often spread over multiple properties in spite of urban main setts typically being smaller than those in rural areas (*Davison et al., 2008*). One third of all cases studied involved a sett with entrance holes on more than one property. Anecdotally, a high level of cooperation between affected neighbours did seem to benefit the success of exclusion efforts in these cases (Supplemental Information 1); co-operation across affected ownership boundaries is required for successful management of wildlife damage (*White & Ward, 2010*).

Badger home ranges in urban England tend to be much smaller than those of their rural counterparts (*Davison et al., 2008*) and movements between social groups, while frequent, may cover shorter distances (*Huck et al., 2008*). Consequently, factors that influence space use and responses to exclusion attempts are likely mainly to exist within the vicinity of

the sett. While it is not possible to discount the importance of un-measured factors in determining the success of exclusion attempts, the most important ones are likely to have been included in the present study.

Measures that might be expected to improve the probability of success when attempting to exclude badgers from a sett include negative stimuli, such as deterrents, and provision of additional resources, such as the construction of an artificial sett. However, badgers have been observed to rapidly habituate to ultrasonic and physical deterrents, rendering them ineffective (*Ward et al., 2008*). Also, as badgers typically have access to multiple setts within their social group range, even in urban environments (*Davison et al., 2008*), it is unclear whether provision of an additional artificial structure at a location not selected by badgers would enhance the likelihood that they would leave their main sett. Moreover, the internal characteristics of artificial setts may be different to those of natural setts (*Kaneko et al., 2013*), making their attractiveness relative to natural setts uncertain.

*Davison et al. (2011)* did not find temporal or spatial clustering of licence applications, and the distribution of cases that we observed is generally consistent with this. Nevertheless, we did observe sequential causation on one occasion, whereby badgers excluded from a sett on one property were the subject of a licence application pertaining to a nearby property. Therefore, we conclude that while successful exclusion of badgers from their setts always moves the badgers to an alternative sett, usually on another property, shifting of the problem between properties, may be infrequent. That is, the presence of badgers on urban properties does not always constitute a problem.

## CONCLUSION

The licensed exclusion of badgers from a problem sett can be an effective means of resolving damage problems. However, successful exclusion depends on a variety of factors, including the type of sett involved and the quality and quantity of the effort invested in excluding badgers from the sett. Licence applicants should be advised that to maximise the chances of successfully excluding badgers all vegetation should be removed from the site, regardless of what type of sett is involved, and that successful exclusion of badgers from a main sett might require substantially more effort than other types of sett.

As urban environments expand and the potential for human-badger conflicts grows throughout Britain, so the need to develop and refine best practice techniques to managing damage problems increases. Further investigation of the behaviour and ecology of urban-dwelling wildlife species that cause problems for humans, and systematic investigation of the effectiveness of existing and novel approaches are required in order to underpin the development of humane, sustainable, environmentally benign, non-lethal solutions.

## ACKNOWLEDGEMENTS

We are grateful to all the Natural England Wildlife Advisors who supplied case information and accompanied us on site visits, and to the homeowners and licensees who granted permission for us to study their cases. We thank Stéphane Pietravalle and Rebecca Callaby for advice on statistical analysis. We are also grateful to Richard Brand-Hardy, Ashley

Matthews, Matt Heydon, Elaine Gill, Rodney Calvert, Simon Mackown, Gary Witmer three anonymous Defra reviewers and two anonymous journal reviewers for helpful comments on drafts of this manuscript.

### Funding

This study was funded by the British Government's Department for Environment, Food and Rural Affairs (Defra) under contracts WM0304 and WM0316. The funding body sought independent peer review of a proposal to undertake this study but were not directly involved in its design or execution. The external funder and their reviewers provided comments on the manuscript before submission. However, the authors were free to respond to these comments as appropriate. The funders had no role in study design, data collection and analysis, decision to publish, or preparation of the manuscript.

### Grant Disclosures

The following grant information was disclosed by the authors:
British Government's Department for Environment, Food and Rural Affairs (Defra): WM0304, WM0316.

### Competing Interests

The authors declare there are no competing interests.

### Author Contributions

- Alastair I. Ward conceived and designed the experiments, performed the experiments, analyzed the data, contributed reagents/materials/analysis tools, wrote the paper, prepared figures and/or tables.
- Jason K. Finney performed the experiments, contributed reagents/materials/analysis tools, reviewed drafts of the paper.
- Sarah E. Beatham performed the experiments, analyzed the data, reviewed drafts of the paper.
- Richard J. Delahay conceived and designed the experiments, contributed reagents/materials/analysis tools, wrote the paper, reviewed drafts of the paper.
- Peter A. Robertson and David P. Cowan conceived and designed the experiments, contributed reagents/materials/analysis tools, reviewed drafts of the paper.

### Animal Ethics

The following information was supplied relating to ethical approvals (i.e., approving body and any reference numbers):

The study proceeded with the approval of the Central Science Laboratory's (now part of the Animal and Plant Health Agency) Ethical Review panel and was conducted under licence 20072352 from Natural England. The broader project within which this study was conducted required the capture and marking of wild badgers under anaesthesia, which was undertaken under licences PPL 60/3351 and PIL 60/9302 from the Home Office.

### Field Study Permissions

The following information was supplied relating to field study approvals (i.e., approving body and any reference numbers):

The capture of badgers was conducted under licence 20072352 from Natural England. The broader project within which this study was conducted required the capture and marking of wild badgers under anaesthesia, which was undertaken under licences PPL 60/3351 and PIL 60/9302 from the Home Office.

### Data Availability

The raw data have been supplied as Supplemental File.

### Supplemental Information

Supplemental information for this article can be found online at http://dx.doi.org/10.7717/peerj.2579#supplemental-information.

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
