# Peer review of "Exclusions for resolving urban badger damage problems: outcomes and consequences"

_PeerJ, doi:10.7717/peerj.2579_

## Round 0.1 · original submission · Minor Revisions

Sorry for the delays - getting referees was difficult.

As you can see though, the reviewers have only minor comments, so assuming you can address these easily the paper should be promptly accepted once returned,

Reviewer 1 ·

Basic reporting

I have put all my concerns/questions in the General comments section below.

Overall I found the article good to read and the results interesting.

Experimental design

No major issues - I have comment on the statistical independence of the repeat cases - see comments below.

Validity of the findings

I think the results are solid however, given the sample size the authors may also need to explain why they think the 32 cases selected are a good representation of the hundreds of sett exclusion that are undertaken each year.

Additional comments

See general comments below:

Line 35: Suggest change to: if a main sett was involved and if vegetation…

Line 53: Suggest add - to the growth of some wildlife…

Line 79: 500-600 – is this a long-term average or best guess? Also what proportion of applications do come from urban areas – 20%.

Line 97: So who applies for the license – is it a land owner or is the license application and control work done by contractors? An explanation would help understanding for overseas readers.

Line 106: Suggest change to – modest-sized sett

Line 131: What type of anesthesia? Is there a standard operating procedure that can be referred too?

Line 207: I recommend using full names for the variable such as “Exclusion type” or “Sett activity”.

Line 219 Suggest change – The remaining variable were all then entered into…

Line 224: It looks like you are doing backwards selection and also using delta AIC values. That is fine but you should reference both techniques.

Line 240: My concern for the two sites studied over two separate years – were the sites independent? By that I mean is it possible that previous attempts at control could influence later success – this needs explaining.

Line 245: OK in Figure 1 - I count 32 locations however, when I add up the cases I get 22 – what am I missing as there were only two repeat cases?

Line 255: Suggest add in sample size of (n=32)

Line 263 Suggest change to 31 days as specified in the methods. Might pay to check throughout.

Line 273: Suggest add only two factors remained….

Line 275: Suggest add level 2 – complete removal…

Line 278: Need to comment on the AIC values – also improvement over the intercept-only model?

Line 279: Suggest add – power of the Final model……

Line 284: Looks like Table formatting has messed up – please correct.

Also I suggest adding lines between each Characteristic break down. That way you can see each section adds up to 32 – makes it easier to follow.

Finally you have included the activity scores – I gather these were dropped from the model as being non-significant? Perhaps worth a comment somewhere that no continuous variable were significant – but are the results of interest? See my comment below regarding the Number of alternative setts.

Line 289: I note you have used Exp(β) – in my stats package it refers to this as the Odds Ratio – might make more sense labeled as such.

Line 294: OK – I have concerns about the sentence starting from: However, the average baseline risk of exclusive failure was low. I am not sure about this statement – the overall baseline was that 12 out of 32 failed – that is still a reasonably high rate at 37.5%. I might be wrong here (I am not a biometrician) but this sentence doesn’t ring true.

Line 320: Suggest add- Chain link was not significantly associated…..

Line 375: Is it just that a main sett is targeted or is it the availability of alternative setts. Table 1 suggests this however, with only 32 cases you may lack statistical power to pick this up. Perhaps a comment re: sample size possible things that could be looked at in future studies.

Line 379: Suggest change sustainable approaches to develop best practice techniques.

·

Basic reporting

Article is fine

Experimental design

Experimental design is fine

Validity of the findings

Findings are valid

Additional comments

This manuscript provides original and useful information. It is well organized and well written. The analyses are appropriate. There is good use of the scientific literature. The figure and tables are appropriate and useful. The supplemental material is interesting reading and gives a better picture of the cases than the analyses alone. A few specific suggestions follow:

I did not see where two references were cited in the text: Gehrt 2004 and Grinder and Krausman 2001. Seems they could have been cited in the Intro, about lines 67-69?
The figure is not very clear/distinct on my copy.
Table 1: the columns need some better aligning.
Lines 33 and 114: use "were" instead of "was"?
Line 347: Cites Davidson et al. 2011, but in the Refs it uses 2011?

Reviewer 3 ·

Basic reporting

Great to see some research addressing the effectiveness of wildlife management strategies and techniques. The context around the research question was very clear. Reporting was generally clear (minor questions below). Most of the references used - particularly in the discussion - focused on badgers, and did not put the research into a broader context for readers. Table 1 needs some work.

Experimental design

This is a very simple study, comparing successful vs unsuccessful badger exclusions and trying to understand the drivers of success and failure. I would suggest the sample size is fairly small in some regard- some justification around the power of the study would be useful. More references around the statistics used.

Validity of the findings

I'm slightly concerned about strength of the conclusion based on the data. Particularly importance of vegetation removal (as compared to the chain link). Sample sizes have come into play here - couldn't directly test some variables.

Additional comments

Great to see some research addressing the effectiveness of wildlife management strategies and techniques. The context around the research question was very clear. Reporting was generally clear (minor questions below). Most of the references used - particularly in the discussion - focused on badgers, and did not put the research into a broader context for readers.

Additional Comments:

1. Ln 70 - more info on status of badgers - are they increasing or declining in urban areas? Currently says ‘well habituated’ - but what is their population status in urban areas?
2. Ln 77 - can you give the percentage increase in urban licences?
3. Sample size should go in methods somewhere - before first mention in results.
4. Ln 212 - ‘colinearity’ should be ‘multicollinearity’ or at very least ‘collinearity’
5. Data analyses - look appropriate, but references (eg Burnham & Anderson) should be used. I would have thought AICc was more relevant given your sample size - corrects for overfitting models where small sample sizes.
6. Results - sites. What is the definition of urban used here? Are all of these licences for urban areas? Or are some rural? Some parameters need to be set around which licences to include for ‘urban setts’.
7. Table 1 could be clearer. A bit of a tidy up perhaps. E.g. perhaps ‘complete’ is in the wrong place eg. frequency rather than ‘number of cases’?
8. AICs not presented (nor Δ w). I know that a second model was run (binary), but shouldn’t the first model be presented? The Wald statistic tends to have some biases when sample sizes are small, so some justification necessary. Some discussion as to the power of the study - with 32 setts used in analyses. The raw numbers don’t suggest the confidence suggested in results & discussion around the importance of vegetation removal.
9. Ln305-308 - can you separate the mechanism of success of veg. removal? Preference by badgers or chain link fence? This seems an important point, but the data is not available to conclude? It appears that the chain link is more important (in your discussion) than the vegetation itself. No mention of chain link in the conclusion at all - when this is why the veg removal might have been successful.
10. Discussion - how urban are these properties that setts can cover more than 1 property? Not a lot of impervious surfaces? More peri-urban? Some context required here. Are there factors contributing to success not measured that are important, such as size of property, connectivity to other non-impervious surfaces, availability of other habitat (eg. distance to nearest park/reserve), etc.
11. What are the other impacts of vegetation removal/disturbance? Negative impacts on stormwater/sedimentation, vegetation itself, etc?

---

## Round 0.2 · accepted · Accept

Alastair

Thanks for your hard work on this. As I said before, it was agony finding referees for this ms

Best wishes